# Fault Detection in the MSW Incineration Process Using Stochastic Configuration Networks and Case-Based Reasoning

**DOI:** 10.3390/s21217356

**Published:** 2021-11-05

**Authors:** Chenxi Ding, Aijun Yan

**Affiliations:** 1Faculty of Information Technology, Beijing University of Technology, Beijing 100124, China; yanaijun@bjut.edu.cn; 2Engineering Research Center of Digital Community, Ministry of Education, Beijing 100124, China; 3Beijing Laboratory for Urban Mass Transit, Beijing 100124, China

**Keywords:** MSW incineration process, fault detection, case-based reasoning, stochastic configuration networks, learning pseudo-metric

## Abstract

Fault detection in the waste incineration process depends on high-temperature image observation and the experience of field maintenance personnel, which is inefficient and can easily cause misjudgment of the fault. In this paper, a fault detection method is proposed by combining stochastic configuration networks (SCNs) and case-based reasoning (CBR). First, a learning pseudo metric method based on SCNs (SCN-LPM) is proposed by training SCN learning models using a training sample set and defined pseudo-metric criteria. Then, the SCN-LPM method is used for the case retrieval stage in CBR to construct the fault detection model based on SCN-CBR, and the structure, algorithmic implementation, and algorithmic steps are given. Finally, the performance is tested using historical data of the MSW incineration process, and the proposed method is compared with typical classification methods, such as a Back Propagation (BP) neural network, a support vector machine, and so on. The results show that this method can effectively improve the accuracy of fault detection and reduce the time complexity of the task and maintain a certain application value.

## 1. Introduction

Municipal solid waste (MSW) incineration has significant advantages in the resourcefulness, harmlessness, and reduction of MSW; therefore, it has become the first choice of MSW treatment technology and has been widely used [1,2]. Because of the high temperature environment in the incinerator, the operation state of the incineration process in the actual industrial site mainly depends on high temperature image observation and the experience of field maintenance personnel for monitoring and fault identification. This method is inefficient and can easily cause a false alarm and misjudgment of the fault. Therefore, it is of great practical significance to develop an accurate and effective fault detection method.

At present, the fault detection methods of the MSW incineration process are mainly divided into model-driven methods [3] and data-driven methods. Due to the strong coupling and non-linearity of the incineration process, it is impossible to obtain an accurate mathematical model. The data-driven method mainly includes a support vector machine (SVM) [4], neural networks (NNs) [5], and multivariate statistical analysis [6,7]. NNs, such as the BP neural network (BPNN) and the radial basis function neural network (RBFNN) are mainly used in the forecasting field [8]. For example, in [9], a transmission line icing prediction model based on a generalized regression neural network and a fruit fly optimization algorithm was proposed, and good prediction results were obtained. In [10], topographic property of a backpropagation artificial neural network was studied, which showed that the best model (high accuracy and no overfitting) of a large-scale BPNN during epochs of training had the lowest local and global efficiency. These results imply that after epochs of training, neurons of the BPNN form local, highly intra-connected functional modules but lack cooperation at a global level. From the above research, we found that NNs have many advantages, but they also have shortcomings, including slow convergence speed and how easily they fall into local optimum and experience-dependent determinations of hidden layers and neuron numbers of networks. In [11], SVM was used for forecasting global incident solar radiation; the results show that approximately 90% of all errors were in the smallest error bracket. In [12], SVM was used for predicting the cost of power transformation projects and achieved good accuracy. In [13], SVM was used for autonomous operation diagnosis of CHP in the microgrid, and a good result was obtained. From the research mentioned above, although SVM has achieved good results, we found that SVM also has shortcomings; for example, it is inefficient for training large scale samples. In multivariate statistical analysis, the variability of the incineration process is not considered, and the assumptions set in practice would limit its application. In recent years, case-based reasoning (CBR), a new machine learning method in the field of artificial intelligence, has attracted wide attention in the field of industrial process fault detection because of its strong learning ability, high problem-solving efficiency, and has the advantage of using past experiences to solve current problems [14,15,16]. However, the similarity measurement method based on Euclidean distance is adopted in the CBR case retrieval process which presents the problem of weight assignment and can easily fall into a distance trap. Therefore, the research of similarity measurement methods has attracted wide attention.

Traditional similarity measurement methods are mainly based on vectors or distance [17]; different measurement methods are usually used for different data types. For qualitative data, vector-based measurements are often used, while for quantitative data, distance-based measurements are often used. The distance measurement method is widely used because of its intuitive and convenient computer physics in practical application [18]. However, when using a distance measurement, such as Euclidean distance [19], if the distance between the solution of the target case and source case is very close, it is considered logical that the solution of the target case can be obtained by referring to the solution of the source case. In some cases, this method is effective and may produce good results in case retrieval [20]. However, in other cases, even if the weight distribution is reasonable and correct, there may be distance traps in the calculation. In other words, the two closest cases are not necessarily the most similar [21]. So far, there are many methods of weight assignment [22,23,24], such as the subjective weighting method and the objective weighting method [25]. The commonly used subjective weighting methods include the expert investigation method, the numerical logic method, the adjustable mean method, etc. [26]. Objective weighting methods mainly include the genetic algorithm (GA) [27], entropy weight (EW) [28], etc. However, the subjective weighting method has much discretionary subjectivity; the characteristic weight distribution based on GA in the objective weighting method can easily fall into local optimization, and the EW method is very sensitive to the variation range of random data. Therefore, it is a challenge to develop an attributive weight assignment method to accurately measure the importance of each attribute in the process of problem solving. In this case, if the distance measure is used as the similarity measure method, the nearest sample cannot be used as the similar sample of the target sample. Therefore, the concept of pseudo metric is introduced based on the definition of metric space, pseudo-metric space [29], the concept of equivalence relation, and equivalence class [30]. Learning pseudo metric (LPM) [31] is when one uses machine learning technology, such as neural networks, to get a pseudo metric and then realizes the similarity of any *x* and *y*. It can avoid the problem of distance measurement. Although LPM can effectively improve the classification accuracy and avoid the difficulty of distance measurement, there are still some drawbacks. One such drawback is that the BP neural network itself can easily fall into local minimum and slow convergence. Therefore, it is necessary to further study the learning pseudo metric.

Based on the above analysis, we propose a SCN-LPM method, and on this basis, we propose a fault detection method based on SCN-CBR in this work. Among them, the SCN-LPM method is obtained by training a SCN model [32], including the construction of a sample set and the definition of pseudo-metric criteria. The fault detection method based on SCN-CBR is obtained by replacing the similarity measurement method based on Euclidean distance with the SCN-LPM method for the case retrieval in CBR, and the validity of the fault detection method based on SCN-CBR proposed in this paper is verified by experiments.

The rest of this paper is organized as follows: After briefly introducing the fault description of the MSW incineration process in Section 2, the SCN-LPM method, the structure and function of the fault detection model based on SCN-CBR for MSW incineration process, fault detection algorithms, and algorithmic steps are given in Section 3. Then, the experiment and result analysis are shown in Section 4. Finally, Section 5 presents our conclusions and future work.

## 2. Fault Description of MSW Incineration Process

The MSW incineration treatment system consists of four key systems: the pretreatment system, the incineration system, the waste heat utilization system, and the flue gas purification system [33,34]. The process flow is shown in Figure 1. After pretreatment, MSW is sent to the incineration system for full combustion through the cross movement of grates at all levels, and the ash produced by incineration is treated by the ash treatment device. The high temperature flue gas generated by incineration is heat recovered through the waste heat utilization system and heated to a steam-driven turbogenerator. Finally, the high temperature flue gas is discharged through the tail flue gas treatment system.

The incineration system in Figure 1 is the key to the whole system. As the backbone equipment of the incineration system, the performance of the waste incinerator directly affects the stable operation of the whole incineration process. Among them, incinerator fault-prone areas are the steam-water system, the horizontal flue gas passage, and the furnace chamber. The types of faults that easily occur in the steam-water system are leakage in the super-heater and economizer. Among these faults, when the super-heater leaks, i.e., the dust accumulation or slagging in the flue is serious, the flue gas temperature will increase, and the flue gas corridor that forms will cause the flue gas flow in the circulation area to increase, thus accelerating the erosion and wear of the super-heater. When the economizer leaks, that is, when a large amount of smoke and dust mixed in the flue gas passes through the flue, it will continuously wear and wash the economizer. Fault types of horizontal flue gas passage that can occur are horizontal flue ash and slagging. In this case, the ash particles in the horizontal flue are in a state of high temperature melting, and the fused slag will cool and condense when it approaches the water wall and eventually enters the flue gas passage with high temperature flue gas to form attachments. Horizontal flue slagging means that the slag adheres to the water wall in a melting state and sticks together to form a hard-to-remove slag layer. The types of furnace chamber faults that can occur are coking and producing slag discharge that is not smooth. In the process of combustion, high-temperature flue gas and ash particles move together. If the furnace chamber temperature is too high, there will be partially melted or semi-melted ash particles, which will form coking when adhered to the heating surface. When slag discharging is not smooth, the large agglomeration of garbage causes a blockage in the discharging outlet. Characteristic variables affecting steam-water system faults, horizontal flue gas passage faults, and furnace chamber faults are shown in Table 1, Table 2 and Table 3, respectively.

## 3. Fault Detection Model for MSW Incineration Process

Aiming at the detection of six kinds of faults in the MSW incineration process, a fault detection model based on SCN-CBR is established in this section. First, a new learning pseudo-metric method is obtained by training an SCN learning model according to a training sample set and defined pseudo-metric criteria. Then, a fault detection model is established based on the problem-solving model of CBR, and the implementation of the algorithm is introduced. Finally, the main algorithm steps are given. However, the implementation of the fault detection method proposed in this paper faces many challenges, such as how to build a SCN-LPM method, how to build a SCN-CBR fault detection model, etc. Next, we conduct in-depth research on these difficult problems and propose solutions.

### 3.1. SCN-LPM Method

(1)Constructing sample set. *C_k_* is a subset of sample set *C* that represents some kinds of real data. *M* is an *n*-dimensional extractor that can map *C_k_* to its characteristic space *F_k_*. Namely:(1)$$F_{k}=\{x=M(e)\in R^{n+1}:e\in C_{k}\},k=1,\ldots ,p$$
where *x* is an element in a point set χ. Sample set *D* is defined as follows:(2)$$D=\{(x,y)\mapsto \delta_{ij}:(x,y)\in F_{i}\times F_{j},i,j=1,\ldots ,p\}$$
where × represents Cartesian product, which can be obtained by combining any two characteristic attributes *F_i_* and *F_j_*. $\delta_{i,j}(x,y)$ represents the Dirichlet symbolic function whose value is 0 when *x* and *y* belong to the same category; otherwise it is 1. That is, when *i* = *j*, *δ_ij_* = 0, otherwise *δ_ij_* = 1. According to the definition of Formula (2), several pairs of sample data can be formed from the sample set and sample set *D* can be constructed. Sample set *D* can be divided into training sample set *D_train_* and testing sample set *D_test_* [31], which can be used to train and validate the following network model. The number of sample data pairs formed is *p*(*p* − 1)/2.(2)Training SCN. The SCN model is trained with a training sample set. The selection of SCNs should consider the structure of the network; that is, the number of nodes in the input layer and output layer as well as the number of neurons in the hidden layer are determined. Document [32] describes the determination method, which is described as follows:

Any given 0 < *r* < 1 and a sequence of nonnegative real numbers $\left\{\mu_{L}\right\}$. Let $\underset{L\to +\infty }{\lim}\mu_{L}=0$, $\mu_{L}\leq (1-r)$. For *L* = 1,2..., Record as:(3)$$\delta_{L}^{}=\sum_{q=1}^{m}\delta_{L,q}^{},\delta_{L,q}^{}=(1-r-\mu_{L}){\left\|e_{L-1}^{},q\right\|}^{2},q=1,2\ldots ,m$$
where *m* is the number of neurons in the hidden layer; $\delta_{L,q}^{}$ is the value of the *q*-*th* neuron in the range of any given *L* hidden layer neurons. ‖.‖ represents the matrix norm and $e_{L-1}^{}{,}_{q}$ is the deviation of the *q*-*th* neuron in the range of *L*-1 hidden layer neurons.

When the number of neurons in the hidden layer is selected, the stochastic function gL needs to satisfy the following inequality:(4)〈eL−1,q,gL〉2≥bg2δL,q,q=1,2,…,m
where gL is a stochastic function (satisfies ∫ℝ|gL(t)|2<∞ or ∫ℝ|g′L(t)|2<∞); Any bias bg∈ℝ+ (positive real field).

The output weights are evaluated as:(5)$$\beta_{L,q}=\frac{\langle e_{L-1,q}{}^{},g_{L}\rangle }{{\left\|g_{L}\right\|}^{2}},q=1,2,\ldots ,m$$

The SCN model randomly assigns input weights and hidden layer node biases under the constraint of inequality (4) and, finally, obtains output. However, for classification problems, it is almost impossible to make the output of the model exactly equal to zero or one when the standard SCN is used. Therefore, the following metrics are used to judge the performance of the model [31]:

(A1) $Y_{NN}(x,y)<\varepsilon_{1}$, When *x* and *y* belong to the same category;

(A2) $Y_{NN}(x,y)\geq \varepsilon_{2}$, When *x* and *y* belong to different categories;

(A3) $|Y_{NN}(x,y)-Y_{NN}(y,x)|\leq \varepsilon_{3}$, For any *x* and *y*;

(A4) $Y_{NN}(x,z)\leq Y_{NN}(x,y)+Y_{NN}(y,z)$, For any *z*, *x* and *y* belong to different categories.

where *x*, *y*, *z* represent eigenvectors; *Y_NN_*(*x*, *y*) is the output of the SCN network model, representing the similarity degree between *x* and *y*; *ε*_1_, *ε*_2_, *ε*_3_ are constants, where *ε*_1_ = *ε*_3_ usually, where the value is 0.2–0.3, and the value of *ε*_2_ is 0.7–0.8.

The termination condition of the model is to satisfy the above metrics (A1–A4) in a certain proportion of α% (α ∈ (0, 100)). For example, when α is equal to 80, *Y_NN_*(*x*, *y*) can be used as the output of similarity degree.

### 3.2. Fault Detection Model Based on SCN-CBR

#### 3.2.1. Model Structure and Function

Figure 2 is a fault detection model based on SCN-CBR, and its main functions are as follows: First, the characteristic attributes of the source case *C_k_* (*k* = 1, 2, …, *p*) and target cases *X_p_*_+1_ are normalized and expressed in the form of eigenvector, and the case base is constructed. Then, the target case *X_p_*_+1_ is used as an input, and the similarity degree between the target case and each source case is measured by the case retrieval method based on SCN-LPM; thus, *K* similar case solutions are obtained. Next, according to the KNN rule, the suggested solution ${\hat{Y}}_{p+1}$ of the target case is obtained. Finally, the suggested solution is confirmed and adjusted through revision, and the corresponding target case and the correct solution *Y_p_*_+1_ is stored in the case base, thus a learning process of reasoning and solving is completed. When the next case arises, the above process is repeated to realize the fault detection function of the MSW incineration process.

#### 3.2.2. Fault Detection Algorithms

The algorithm implementation of each part shown in Figure 2 is described below.

(1)Constructing case base. The problem descriptions and solutions of target case *X_p_*_+1_ and source case *C_k_* are normalized and expressed as an eigenvector form in binary tuples to form *p* source cases, which are stored in the case base. Each source case is recorded as $C_{k}(k=1,\;2,\ldots ,p)$. It can be expressed in the form of binary tuples as follows:(6)$$C_{k}:\langle X_{k};Y_{k}\rangle ,k=1,2,\cdots p$$
where *p* is the total number of source cases, *X_k_* is the set of characteristic attributes in the *k-th* source cases, and *Y_k_* is the category of characteristic attributes in the *k-th* source cases. Assuming that each source case has *n* characteristic attributes, *X_k_* can be expressed in the following form:(7)$$X_{k}=(x_{1,k},\cdots ,x_{i,k},\cdots ,x_{n,k})$$
where *x_i_*_,*k*_ is the normalized value of the *i*-*th* characteristic attribute in the record of article *k-th.*(2)Case retrieval based on SCN-LPM. The input variable of the target case *X_p_*_+1_ and the input variable of the source case *X_k_*(*k* = 1, 2, …, *p*) are composed of *p* input pairs, namely:

(8)
$$D_{k}:\langle X_{p+1};X_{k}\rangle ,k=1,2,\cdots p$$

Then, *p Y_NN_*(*X_p_*_+1_, *X_k_*) can be obtained according to the SCN-LPM method. According to A1 of the four metrics in Section 3.1, *K* source cases similar to the target case *X_p_*_+1_ can be obtained.(3)Case reuse. According to the KNN rule, the number of categories corresponding to the *K* source cases retrieved is counted, and the category with a large number is taken as the suggested category ${\hat{Y}}_{p+1}$.(4)Case revision. When evaluating the suggested category ${\hat{Y}}_{p+1}$, if the evaluation is unsuccessful, the classification results need to be revised to obtain the correct category ${\hat{Y}}_{p+1}$.(5)Case storage. Target case and corrected category are stored in the case base to form a new case. So far, the number of source cases has been from *p*→*p*+1, and the CBR problem solving process is completed.

### 3.3. Algorithmic Steps

In summary, the steps of the SCN-LPM algorithm are as follows:

Step 1: Normalize the characteristic attributes in the sample set and perform the initialization of parameters, setting SCN learning parameters and stochastic parameters, etc.

Step 2: Construct sample set *D* according to Formula (2) and divide it into training set *D_train_* and testing set *D_test_*.

Step 3: Determine the structure of the SCN model, then randomly assign input weight and hidden node bias under the constraint of inequality (4), so as to determine the number of input and output nodes as well as the number of hidden layer neurons, i.e., determine the structure of SCNs.

Step 4: Train SCNs with training set *D_train_*.

Step 5: Determine whether the SCNs model satisfies the measurement criterion (A1–A4) in a set proportion of a%. If it satisfies the measurement criterion, continue to Step 6; otherwise, return to Step 4.

Step 6: Taking the target case as input, the similarity degree between the target case and each source case is measured through the SCN-LPM method, and the similarity degree $Y_{NN}(x,y)$ is the output. According to A1 of the metrics, *K* similar case solutions can be obtained.

## 4. Experimental Study

For easy viewing, the abbreviations are specified as follows: the support vector machine algorithm is recorded as SVM, the Euclidean distance similarity measure method is recorded as E, the BP-based learning pseudo metric is recorded as BP-LPM, the SCN-based learning pseudo metric method is recorded as SCN-LP, the CBR algorithm based on the Euclidean distance similarity measure is recorded as ED-CBR, the CBR algorithm based on BP-LPM is recorded as BP-CBR, and the CBR algorithm based on SCN-LPM is recorded as SCN-CBR.

### 4.1. Experimental Parameters

The experiment is programmed in a MATLAB R2016a 9.0.0 environment. The computer CPU used is Intel (R) Core (TM) i5-4570H CPU@3.20 GHz, and the memory is 8 GB.

The parameters of various methods are as follows: In the SVM algorithm, the penalty factor is 10, the loss factor is 0.1, and the kernel function is the Gauss radial basis function. In the BP algorithm, a three-layer network structure is adopted, the number of hidden layer neurons is 15, the activation function is a Sigmoid function, the training function is Trainrp, training times are 1200, the value of the target error is 10^−5^, and the learning rate and convergence error are 0.1 and 0.01, respectively. In the ED-CBR algorithm, the value of K is 5. In the BP-CBR algorithm, *ε*_1_ = *ε*_3_ = 0.3, *ε*_2_ = 0.7, and the condition of terminating the LPM model is to satisfy the proportion of the measurement criterion (A1–A4) in Section 3.2 which is 80%; the other parameters are the same as those of BP algorithm. In the SCN-CBR algorithm, *ε*_1_ = *ε*_3_ = 0.3, *ε*_2_ = 0.7, the condition of terminating the LPM model is to satisfy the proportion of the measurement criterion (A1–A4) in Section 3.2 which is 80%, the three-layer network structure is adopted, the maximum number of hidden layer nodes is 100, the training error limit is 0.01, the range of stochastic weight is (0.5, 1, 5, 10, 30, 50, and 100), the maximum stochastic configuration time is 100, and the activation function is a Sigmoid-type function. All parameters are obtained by a grid search method and a 10-fold cross-validation method. That is, the possible values of various parameters are arranged and combined, and all possible combination results are listed to generate a “grid”. Then, each combination is used for model training, and the effect is evaluated by a 10-fold cross-validation. After all the parameter combinations are tried, it returns an appropriate model and automatically adjusts to the best parameter combination.

The experimental data are collected from a DCS-distributed control system of a MSW incineration power plant in Beijing, sampling every 5 min. The sample size of the steam-water system is 390, the sample size of horizontal flue gas passage is 345, and the sample size of the furnace chamber is 409. Sample set *D* is constructed and divided into training sample set *D_train_* and testing sample set *D_test_* according to Section 3.1. Among them, nine pieces are selected as training sets and one piece is selected as a testing set.

### 4.2. Performance Testing

#### 4.2.1. Stability

In order to evaluate the stability of SCN-CBR model, a stability experiment was carried out with steam-water system fault data as an example. The specific process is as follows: First, the dataset was selected as a case base, sample set D was constructed, and 1000, 4000, 8000, 12,000, 16,000, and 20,000 input pairs were randomly selected from training set *D_train_* and testing set *D_test_*, and were consequently separated from sample set. Then, they were put into the pseudo-metric model and results were produced. Finally, the satisfaction rate of the measurement criteria A1-A4 was verified and shown in Table 4. It can be seen from the table that with the increase of the number of input pairs, the standard deviations of the training set satisfying the measurement criteria are 0.59, 0.39, 0.37, and 0.45, respectively, and the standard deviations of the testing set satisfying the measurement criteria are 0.69, 0.46, 0.35, and 0.54, respectively, which indicates that the satisfaction rates of the training set and the testing set are basically unchanged. The coefficient variation (CV) of the training set satisfying the measurement criteria are 0.61%, 0.41%, 0.41%, and 0.51%, respectively, and the coefficient variation (CV) of the testing set satisfying the measurement criteria are 0.72%, 0.52%, 0.37%, and 0.63%, respectively. We can see that all the coefficient variations (CV) are less than 1%, which indicates that the satisfaction rate has weak variation. That is to say, the SCN-CBR model can control the dataset within a reasonable accuracy range and has good stability.

#### 4.2.2. Robustness

In the presence of noise, the robustness of six kinds of faults (fault 1: super-heater leakage, fault 2: economizer leakage, fault 3: horizontal flue ash accumulation, fault 4: horizontal flue slagging, fault 5: furnace chamber coking, fault 6: slagging is not smooth) was tested on the corresponding datasets. First, each fold cross validation experiment automatically produced a stochastic vector which obeys the uniform distribution of (−1, 1), i.e., noise *noise**_i_*. Then, 10 different interference factors *λ_i_* (*i* = 1, 2, …, 10) were added to the noise. Thus, the input vectors of the SCN-CBR model became the sum of the input vector and noise, i.e., [*I* + *λ_i_* × *diag*(*noise**_i_*)]*s*, where *λ_i_* changes from 1% to 10%. *I* is a unit matrix with compatible size and *diag*(·) is the diagonal matrix. In the presence of different interference factors and noises, the change curve of classification accuracy of the fault detection model is shown in Figure 3. The graph shows that the classification accuracy of the model does not fluctuate obviously after adding different interference factors, which shows that the fault detection model has a certain anti-interference ability and good robustness.

The result statistics of the classification accuracy rate of the fault detection model under the action of different interference factors are shown in Table 5. From the table, we can see all the coefficient variations (CV) are less than 1%, which indicates the six faults have weak variation under the action of different disturbance factors. That is to say, the fault detection model has good anti-interference ability and good robustness.

### 4.3. Contrast Experiments

In order to further verify the validity of the fault detection model, the proposed method was compared with BP, SVM, ED-CBR, and BP-CBR by using fault data, and the accuracy of fault detection is given. All selected parameters are obtained using the grid search method and 10-fold cross-validation method, which make the models have the best effects. In the case of best parameters, the comparison experiment of single fault-detection accuracy and multi-fault detection accuracy are carried; the results are shown in Figure 4 and Figure 5, respectively.

Figure 4 shows that the classification accuracy of the fault detection model based on SCN-CBR is the highest in six different faults of three different regions, which indicates that the single fault-detection accuracy of the proposed method has a strong advantage.

Figure 5 shows that the detection accuracy of the proposed method is higher than that of other methods in various faults of each region. In summary, the fault detection method based on SCN-CBR has a better comprehensive performance in the process of MSW incineration treatment.

As for the running time of different fault detection methods for the steam-water system, the horizontal flue gas passage, and the furnace chamber, it can be seen from Figure 6, Figure 7 and Figure 8 that SCN-CBR has a longer running time than BP, SVM, and ED-CBR, but a shorter running time than BP-CBR. This shows that SCN-CBR can effectively improve the BP-CBR model which has a long training time with the increase of sample size. In terms of the detection accuracy and running time, the comprehensive performance of SCN-CBR is more advantageous.

## 5. Conclusions

In order to obtain a fault detection method in the waste incineration process, a learning pseudo metric method based on SCN was studied, and a fault detection model based on SCN-CBR was designed. The experiment was carried out through the historical data of a waste incineration plant, and the results show that the method proposed in this paper has higher accuracy in fault detection. The main contents are as follows:
(1)A learning pseudo metric method based on SCN was constructed. First, the sample set was constructed according to the Cartesian product. Then, the pseudo metric criterion was defined. Finally, according to the training sample set and the defined pseudo metric criteria, the SCN learning model was trained, and a new learning pseudo metric method was obtained.(2)A fault detection model based on SCN-CBR was constructed. The similarity measurement method based on SCN-LPM was applied to the retrieval stage of CBR, and a fault detection model of the waste incineration process based on SCN-CBR was established.


Using the SCN-LPM similarity measurement method instead of distance measurement could effectively avoid the problems of weight assignment and distance trap, as well as the fast-learning ability and the universal approximation property of utilized SCNs, which could effectively improve local minimization and slow convergence speed of BP-LPM, and could also reduce the detection time complexity. At the same time, the incremental learning of CBR was also utilized to store the experience of solved problems to improve the accuracy of subsequent problem solving. However, there are still some limitations. For example, due to the inherent characteristics of SCNs, the network overfitting phenomenon may occur, resulting in low generalization ability. Besides, the instability of samples may lead to inaccurate fault detection. Thus, future work can be carried out from two aspects. One aspect is how to improve the network structure of SCN and optimize SCN parameters, and another is how to ensure the stability of sample data.

## Figures and Tables

**Figure 1 sensors-21-07356-f001:**
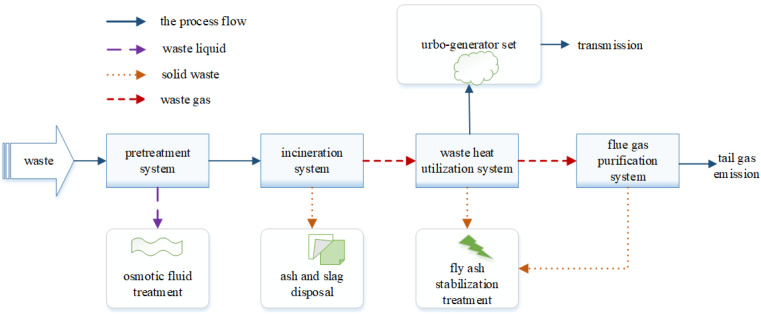
MSW incineration process flow.

**Figure 2 sensors-21-07356-f002:**
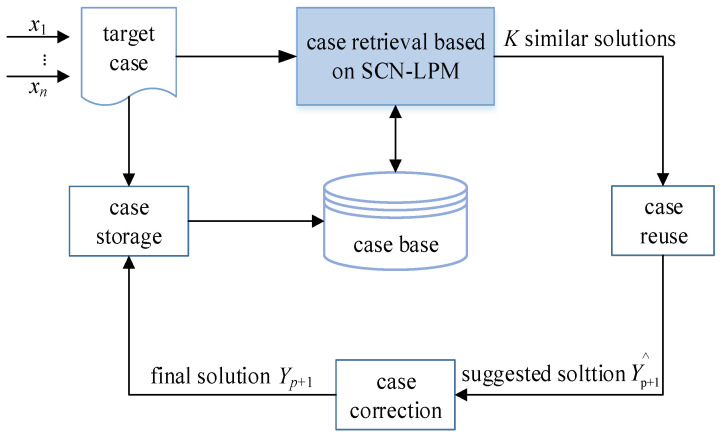
Fault detection model based on SCN-CBR.

**Figure 3 sensors-21-07356-f003:**
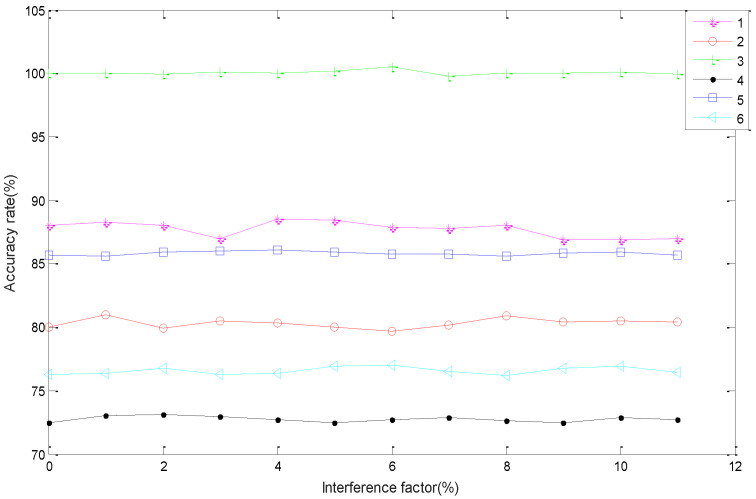
Robustness test.

**Figure 4 sensors-21-07356-f004:**
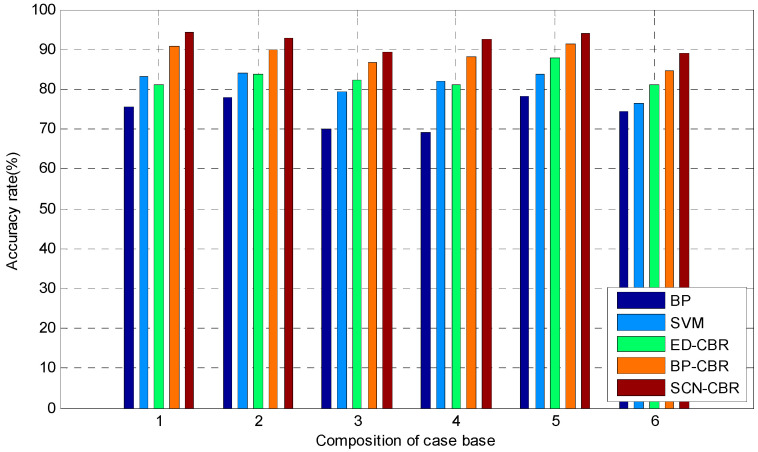
Single fault detection accuracy (%).

**Figure 5 sensors-21-07356-f005:**
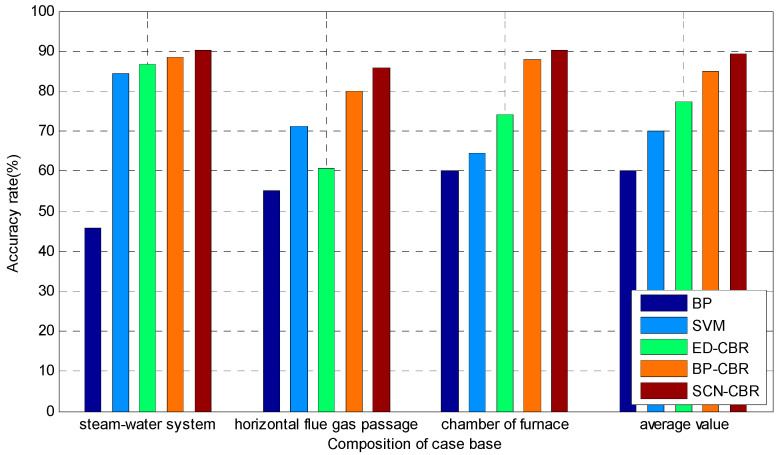
Multi-fault detection accuracy (%).

**Figure 6 sensors-21-07356-f006:**
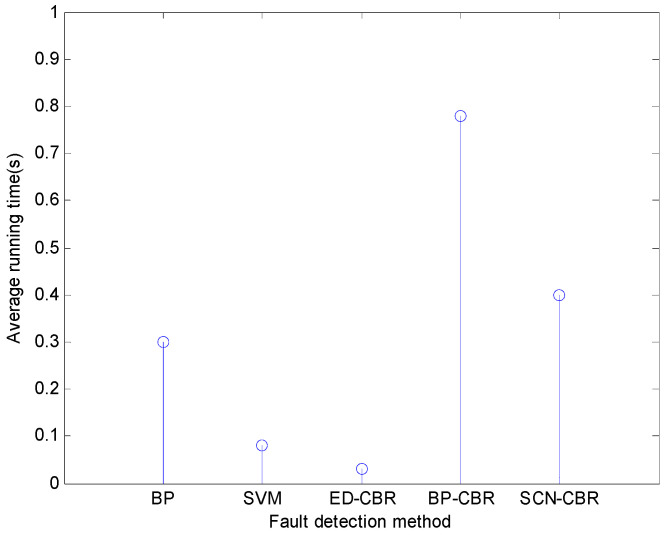
The running time of different methods for steam-water system fault detection (*s*).

**Figure 7 sensors-21-07356-f007:**
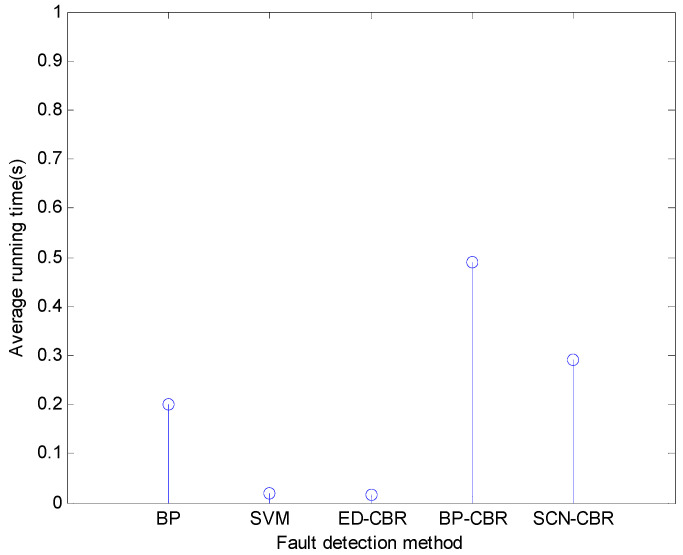
The running time of different methods for horizontal flue gas passage fault detection (*s*).

**Figure 8 sensors-21-07356-f008:**
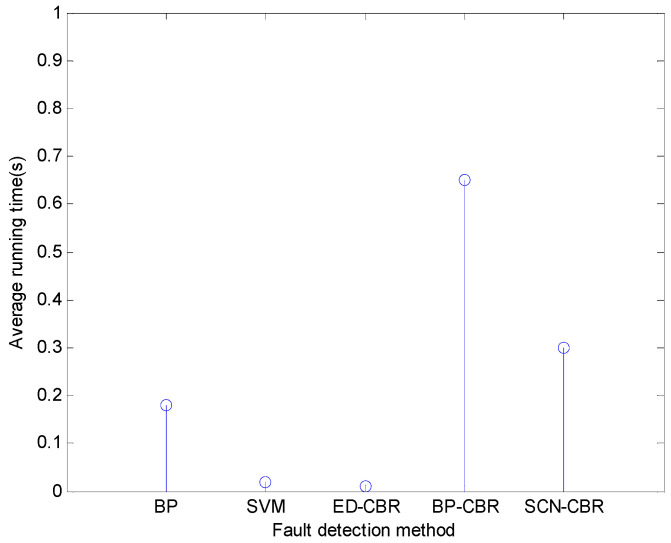
The running time of different methods for chamber of furnace fault detection (*s*).

**Table 1 sensors-21-07356-t001:** Faults and influencing factors of steam-water system.

Serial Number	Fault Type	Influence Factor
1	Leakage of super-heater	boiler drum water level *x*_1_, feed pump outlet total flow *x*_2_, primary and secondary super-heater cooling water flow *x*_3_, boiler outlet main steam flow *x*_4_, three-stage super-heater inlet flue gas temperature *x*_6_, three-stage super-heater outlet steam pressure *x*_7_, protective pipe inlet flue gas temperature *x*_8_, evaporator inlet flue gas temperature *x*_9_.
2	Leakage of economizer	

**Table 2 sensors-21-07356-t002:** Faults and influencing factors of horizontal flue gas channel.

Serial Number	Fault Type	Influence Factor
1	Horizontal flue ash deposit	boiler outlet main steam flow *x*_4_, furnace negative pressure *x*_5_, three-stage super-heater inlet flue gas temperature *x*_6_, protective pipe inlet flue gas temperature *x*_8_, evaporator inlet flue gas temperature *x*_9_, flue gas temperature of economizer import *x*_10_.
2	Slagging in horizontal flue	

**Table 3 sensors-21-07356-t003:** Furnace faults and influencing factors.

Serial Number	Fault Type	Influence Factor
1	Furnace coking	furnace temperature *x*_11_, air flow rate of grate in drying section *x*_12_, air flow rate of grate in combustion section I *x*_13_, air flow rate of grate in combustion section II *x*_14_, air flow rate of grate in burning section *x*_15_, secondary air flow *x*_16_, exit flue gas temperature of economizer *x*_17._
2	Slagging discharge is not smooth	

**Table 4 sensors-21-07356-t004:** Satisfaction rate of measuring criteria for steam-water systems (%).

Input Pairs	Training Set				Testing Set			
	(A1)	(A2)	(A3)	(A4)	(A1)	(A2)	(A3)	(A4)
1000	96.57	95.24	91.30	89.34	96.34	89.02	95.23	86.24
4000	97.32	96.02	92.05	89.57	97.48	88.53	95.17	85.49
8000	96.81	95.67	91.84	89.25	96.01	89.74	95.46	86.02
12,000	97.29	95.38	91.79	88.36	96.27	88.36	95.12	87.10
16,000	95.75	96.35	90.96	89.02	95.51	88.68	96.04	86.28
20,000	96.04	95.39	91.38	88.43	97.30	89.12	95.85	86.93
Average value	96.63	95.68	91.55	89.00	96.49	88.91	95.48	86.34
Standard deviation	0.59	0.39	0.37	0.45	0.69	0.46	0.35	0.54
Coefficient variation	0.61%	0.41%	0.41%	0.51%	0.72%	0.52%	0.37%	0.63%

**Table 5 sensors-21-07356-t005:** Accuracy rate (%).

Interference Factors	Classification Accuracy Rate					
	Fault 1	Fault 2	Fault 3	Fault 4	Fault 5	Fault 6
1	88.30	81.02	99.89	73.09	85.61	76.41
2	88.01	79.90	99.87	73.11	85.90	76.82
3	87.02	80.49	99.91	72.98	85.99	76.30
4	88.52	80.32	99.74	72.71	86.11	76.41
5	88.41	80.02	99.93	72.52	85.89	76.90
6	87.89	79.77	99.96	72.70	85.78	77.01
7	87.78	80.19	99.81	72.92	85.78	76.52
8	88.01	80.91	99.83	72.65	85.62	76.20
9	86.90	80.43	99.85	72.53	85.81	76.80
10	86.90	80.50	99.87	72.90	85.89	76.89
Average value	87.77	80.36	99.87	72.81	85.84	76.63
Standard deviation	0.62	0.41	0.06	0.22	0.15	0.29
Coefficient variation	0.7%	0.5%	0.06%	0.3%	0.1%	0.4%

## Data Availability

The data presented in this study are available on request from the corresponding author. The data are not publicly available due to privacy.

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
