# Peer review of "Fault Detection in the MSW Incineration Process Using Stochastic Configuration Networks and Case-Based Reasoning"

_sensors, 2021, doi:10.3390/s21217356_

Round 1

Reviewer 1 Report

Aiming at the difficult problem of fault detection accuracy in municipal solid waste incineration process, a fault detection method for waste incineration process was proposed by combining stochastic configuration networks (SCNs) and case-based reasoning (CBR) in this paper, the structure and function of fault detection model were given, the algorithm implementation was discussed, and the method was analyzed and verified by the historical data of a waste incineration plant. This paper has good theoretical significance and application value. The deficiency is that the English expression of the paper still needs to be corrected. 

Author Response

Reply: Thanks for your recommendation and comment. We have corrected the English expression of the paper. The modifications are as follows:

Section 1:

① Replace “…and has become the first choice of MSW treatment technology…” by “…so it has become the first choice of MSW treatment technology…”

② Replace “…the high temperature environment of the incinerator…” by “…the high temperature environment in the incinerator…”

③ Replace “…the research of similarity measurement method…” by “…the research of similarity measurement methods…”

Section 3:

  • Replace “…that represents some kind of real data…” by “…that represents some kinds of real data…
  • Replace “Then, The target case…” by “Then, the target case…”
  • Replace “So far, the number of source cases is from pp+1…” by “So far, the number of source cases has been from pp+1…”

Reviewer 2 Report

Aiming at the fault detection of municipal solid waste (MSW) incineration process, this paper combines stochastic configuration networks (SCNs) and case-based reasoning (CBR) to establish an SCN-CBR fault detection model. The similarity measure method based on learning pseudo-metric instead of distance metric can effectively avoid the problems of weight assignment and distance trap. By fast learning ability and universal approximation property of SCNs, local minimization can effectively be avoided, and pseudo-metric-based similarity measures in SCNs can also reduce detection time complexity so as to show faster convergence than distance metric-based similarity. At the same time, the incremental learning of CBR is also utilized to store the experience of solved problems so as to improve the accuracy of subsequent problem-solving. The experimental results show that the SCN-CBR model has certain robustness and stability, and has good learning ability and solving performance compared with other fault detection models. The method provided in this paper has application potential in smart cities and sustainable social development. The paper also has the following problems to be revised.

  • What are the difficulties in the implementation of the fault detection method proposed in this paper? This point needs to be explained in the text.
  • In Section 3.1, Xi and Xj in “which can be obtained by combining any two characteristic attributes Xi and Xj” should be consistent with Fi and Fj in equation (2).
  • Abbreviations should be given in full when they first appear, such as BP.
  • In Section 3.1, (A1), y in “(A1), When x and y belong to the same category” should be italicized. There are similar problems in other places, please modify them together.
  • The authors need to correct existing grammar mistakes.

Reviewer 3 Report

General Comment:

Overall, the paper provides an interesting and promising methodology, which however needs to be more clearly articulated in certain critical regards. There is therefore a set of recommendations offered below, some fo which I consider critical for the paper to be accepted.  

Specific comments and recommendations:

In the introduction, there is criticism in general of NNs and SVMs and more specific literature should be provided; efficiency is dependent on the type of problem targeted and that needs to be accurately reflected

On your discussion of uses of Euclidean distance: “usually assumes that the weights of attributes are equal and the attributes are independent of each other “ - is this accurate? I am sure I’ve come across cases with advanced weighting approaches. Please provide literature.  

There is an LPM reference dating back to 2005; there is more recent literature using the method which may be worth acknowledging and reviewing. 

Similarly in the Intro, and while the SCN-LPM approach is provided, there is little discussion of recent literature on the combinatory use of SCN-driven approaches with CBR. A more detailed review needs to be provided here. 

Directly relevant to the above, there seems to be a need to outline the novelty of the paper,  which seems to be evasive. Possibly providing a set of bullet point crystallising the authors’ contribution to the literature.  

Sections 2-4 provide a satisfactory explanation of the approach, with only minor editorial and style observations (e.g. spacing and indentation of 3.1 and 3.3) - otherwise, a complete and accurate outline of the approach.  

In 4.1, where the experimental parameters are provided, these are not in any way qualified. It is understood that not a full variety of parameters can be tested, however, if there has been a selection and parameter tuning methodology that led to the values provided, that has to be explained. That observation applies across the board, including the selection of the 10-fold cross validation used for performance measurement (I.e. provide a justification with reference to literature). 

In Table 4, please highlight (bold) the values of  significance for your results. Also you mention “has certain stability”. It is not entirely clear what would be the threshold you would use to characterise the approach unstable, therefore this needs to be explained. You will need to connect that statement with the figures mentioned immediately above. For instance, you claim these are “basically unchanged”, but that statement needs further qualification. 

A similar observation on Robustness - the statement reads: “Fig. 3. The graph shows that the classification accuracy of the model does not fluctuate obviously after adding different interference factors..” - again, there needs to be a reference metric that would be used for robustness (as above, a threshold). Possibly a table demonstrating fluctuation levels would be useful. 

In the final comparison, where the recommended approach appears to  consistence outperform the others tested against, there is again the need to underline how you have ensured that parameterisation of the competing approaches ensures comparability. This links back to my comment on section 4.1. This is critical, as it will play a major role for ascertaining your claim. 

In section 5, please provide more clear conclusions (perhaps signposted as bullet points) as well as a more extensive discussion fo your limitations and future work. 

Round 2

Reviewer 3 Report

Only observations requiring attention are provided below. The remaining points have been satisfactorily addressed. Points needing attention will have to be addressed as they affect the articulation of results comparability and the experimental design.   

Minor correction: In the introduction, change “In reference [X] “ to “In [X] “

More important corrections:

(1)

REVIEWER: Similarly in the Intro, and while the SCN-LPM approach is provided, there is little discussion of recent literature on the combinatory use of SCN-driven approaches with CBR. A more detailed review needs to be provided here.

AUTHORS' REPLY: Thanks for your kind comment. We don’t provide more detailed review on the combinatory use of SCN-driven approaches with CBR. The main explain is as follows: The combinatory use of SCN-driven approaches with CBR is proposed in our paper, and it is one of the novelty of our paper, so there is little discussion of recent literature on that.

NEW REVIEWER COMMENT:

>> There is then a case for stating the novelty in more clarity. As already proposed, you may want to signpost your contribution and novelty at the end of the introduction as well as in the abstract, making clear what parts of your proposal are novel. 

(2)

NEW REVIEWER COMMENT:

>> In 4.1 on experimental parameters: - the response provided does not address my original point: what is needed here is the research design rationale that led to the specific selections and reach the conversion point; as also mentioned, my request applies across the board - how do you know, for instance, that a finer parameter tuning could not be done to optimise further?

(3) 

REVIEWER: In the final comparison, where the recommended approach appears to consistently outperform the others tested against, there is again the need to underline how you have ensured that parameterisation of the competing approaches ensures comparability. This links back to my comment on section 4.1. This is critical, as it will play a major role for ascertaining your claim.

AUTHORS' REPLY: Thanks for your kind suggestion. We have underlined again how we have ensured that parameterisation of the competing approaches ensures comparability. Pleased see the first paragraph in section 4.3. The main statement is as follows:

“…All selected parameters are corresponding to the best results through repeated experiments with different parameters on the test data set...”

NEW REVIEWER COMMENT:

>> the same comment as the one I provided on 4.1 applies. There is more to be explained here to address how have you ensured that you are not comparing, say, an inferiorly parameterised version of SVM with an optimised SCN-CBR? For instance, one way of securing relatively comparable results is to ensure that the configuration of all models was optimised for the specific experimental setting.
